# Pigment of *Ceiba speciosa* (A. St.-Hil.) Flowers: Separation, Extraction, Purification and Antioxidant Activity

**DOI:** 10.3390/molecules27113555

**Published:** 2022-05-31

**Authors:** Boyu Chen, Afzal Misrani, Cheng Long, Zhizhou He, Kun Chen, Li Yang

**Affiliations:** 1Precise Genome Engineering Center, School of Life Sciences, Guangzhou University, Guangzhou 510006, China; chenboyu875@163.com; 2South China Normal University-Panyu Central Hospital Joint Laboratory of Translational Medical Research, Panyu Central Hospital, Guangzhou 511400, China; afzal_misrani@gzhu.edu.cn (A.M.); longcheng@m.scnu.edu.cn (C.L.); 3School of Life Sciences, South China Normal University, Guangzhou 510631, China; 4Chemistry and Chemical Engineering, Guangzhou University, Guangzhou 510006, China; 5School of Life Sciences, Guangzhou University, Guangzhou 510006, China

**Keywords:** *Ceiba speciosa*, pigment, extraction, purification, lipopolysaccharide, antioxidant

## Abstract

In this work, the extraction procedure of a natural pigment from the flower of *Ceiba speciosa* (A. St.-Hil.) was optimized by response surface methodology. It is the first time that the extraction of the flower pigment of *C. speciosa* (FPCS) has been reported, along with an evaluation of its stability and biological activity under various conditions, and an exploration of its potential use as a food additive and in medicine. Specifically, the effects of ethanol concentration, solid–liquid ratio, temperature and time on the extraction rate of FPCS were determined using a Box–Behnken design. The optimum extraction conditions for FPCS were 75% ethanol with a solid–liquid ratio of 1:75 mg/mL) at 66 °C for 39 min. The purification of FPCS using different macroporous resins showed that D101 performed best when the initial mass concentration of the injection solution was 1.50 mg/mL, resulting in a three-fold increase in color value. The yield of dry flowers was 9.75% of fresh petals and the FPCS extraction efficiency was 43.2%. The effects of light, solubility, pH, temperature, sweeteners, edible acids, redox agents, preservatives and metal ions on FPCS were also investigated. Furthermore, the characteristics of FPCS were determined by spectrophotometry at a specific wavelength using the Lambert–Beer law to correlate the mass of FPCS with its absorbance value. An acute toxicological test performed according to Horne’s method showed that FPCS is a non-toxic extract and thus may be used as a food additive or in other ingestible forms. Finally, western blotting showed that FPCS prevents lipopolysaccharide-induced hippocampal oxidative stress in mice. The study suggests that FPCS may function as an antioxidant with applications in the food, cosmetics and polymer industries.

## 1. Introduction

Nature has long been recognized as a source of innumerable biologically active compounds, many of which have medical applications either as purely natural products or as their derivatives [1,2,3,4]. However, the limited availability of source material for natural products such as pigments can hamper biological studies and the clinical development of interesting compounds. In this context, both industry and academia are continuing to develop sustainable methods of extraction involving technologies that can isolate biologically active natural products quickly and cheaply [5,6,7].

Edible natural pigments can be obtained from natural resources including animals, plants and microorganisms. Natural pigments, unlike synthetic pigments, are generally safe and non-toxic. As well as their natural coloring properties, most of these pigments have some pharmacological value, either alone or in combination with other compounds, which has provoked recent attention from researchers and industrialists [8,9].

*Ceiba speciosa*, also known as the floss silk tree, is a large tree within the Bombacaceae family. It occurs in tropical and subtropical forests in South American countries such as Argentina, Bolivia, Brazil, Peru and Paraguay [10,11], and is also found in Guangdong province, China. It is also consumed in Northern and Eastern Nigeria as a traditional medicine for diabetic control. This tree is also found in Palermo city, representing its availablity in Europe [12]. *C. speciosa* is a fast-growing deciduous tree with unique and appealing characteristics. It produces soft compound leaves with six to eight leaflets, while its large, ornamental flowers are pale pink with five frilly petals and a cream-colored or yellow center. This tree has been investigated on several levels. Thus, a complex polymer with a backbone made up of glucuronosyl and mannosyl units has been found in the gum exudate, which is exuded when the trunk is injured [11]. *C. speciosa* fruit elicits anti-inflammatory and antioxidant responses via phenolic compounds that inhibit hydrogen peroxide-induced mitochondrial membrane depolarization and caspase-9 activation [11,13]. Preliminary research suggests that *C. speciosa* leaf extracts have antioxidant, anti-inflammatory, antipyretic, antibacterial and antifungal functions [14,15], while its seed oil shows hypoglycemic and anti-obesity properties [16]. An aqueous extract of *C. speciosa* bark has a low toxicity and the potential to treat hyperglycemia in animals [17]. However, to date, there has been no investigation of the extraction and possible beneficial functions of the flower pigment of *C. speciosa* (FPCS).

The aim of the present work was to determine and validate optimum extraction conditions for a quantitative analysis of FPCS and to further explore the potential of FPCS as an antioxidant and as a new source of natural pigment. Several parameters were evaluated, such as the solvent concentration, solid/liquid (SL) ratio, extraction time, ultrasound power and extraction temperature, that may significantly affect the extraction efficiency. The study showed that some parameters, such as the volume fraction of ethanol, SL ratio and homogenization time, were critical for the yield and purity of FPCS. Finally, the results of bioactivity screening suggest that FPCS is non-toxic and may function as an antioxidant.

## 2. Materials and Methods

### 2.1. Extract Preparation

FPSC was harvested from October to December 2020 from different *C. speciosa* trees in Guangzhou (23°7′0′′ N 113°15′ E), Guangdong Province, China. The species and its flowers were identified by a botanist. In particular, the gathering of flowers was carried out in the city by collecting them randomly from 30 healthy plants. The collected flowers were deposited in the School of Life Sciences (FCSBC10122020) and delivered to a laboratory, washed with distilled water and then air dried for about a week. After cleaning and drying, the yield of dry flowers was 9.75% of fresh petals. Dry flowers were crushed with a grinder, passed through an 80-mesh sieve and stored at 4 °C. Ethanol was used for extraction since it had a higher extraction efficiency than water in our preliminary experiments.

### 2.2. Response Surface Methodology (RSM)

Lixiviating and Thermal baths were used to perform the extraction. The study used the experimental design of pigment extraction which mainly considers four factors: extractant concentration, solid–liquid ratio, temperature and extraction time. Based on the verification of a single-factor experiment, all four factors had a significant impact on the extraction rate, indicating that the extraction rate was affected by multi-factors and should be analyzed by RSM [18]. The Box–Behnken design (BBD) was selected to obtain the best extraction conditions when there are more than three factors which act together. The ranges of four factors: extractant concentration (65–85%), material–liquid ratio (1:60–1:80), temperature (60–80 °C) and time (45–60 min).

The process flow was as follows: petal harvesting → cleaning → drying → crushing → ethanol solution extraction → centrifugal separation → filtration → rotary evaporation → drying → crude product.

The *C. speciosa* petal powder was accurately weighed into a triangular flask and the crude product was extracted according to the process flow. After accurately weighing the quantity of the crude product, the yield of crude pigment was calculated according to the following formula:Yield (y) = mass of crude pigment/mass of petal powder × 100%

### 2.3. Purification of FPCS

Macroporous resin is a styrene-type non-polar copolymer that was used as an adsorbent to remove impurities such as colloids, starch, sugar, fat, tannin and inorganic salts in the crude FPCS preparation. The porosity of D101 macroporous resin is 42–46%, with an averaged pore size of 25–28 nm and granulation (particle size range 0.3–1.25 nm) ≥ 95%. The porosity of A8 macroporous resin is 42–46%, with an averaged pore size of 12–16 nm and granulation (particle size range 0.3–1.25 nm) ≥ 95%. The porosity of HPD100 macroporous resin is 42–46%, with an averaged pore size of 8.5–9 nm and granulation (particle size range 0.3–1.25 nm) ≥ 95%. All resins were from Donghong Chemical Co., Ltd. (Dongguan, China). A static adsorption method was used: the activated macroporous resin was accurately weighed into a conical flask and an FPCS solution of the same volume and concentration was added; the flask was shaken at room temperature (100 rpm) for 24 h; then, the absorbance (A_1_) of the supernatant at the maximum absorption wavelength was measured every 2 h and the static adsorption rate and adsorption capacity were calculated as follows:Adsorption rate = (A_0_ − A_1_)/A_0_ × 100%;
Adsorption capacity = (A_0_ − A_1_) × V_0_/M
where M is the mass of resin; A_0_ is the absorbance value of pigment stock solution; V_0_ is the volume of pigment stock solution; and A_1_ is the absorbance value of thesupernatant after adsorption.

For the dynamic adsorption method, the activated macroporous resin was placed in a chromatographic column and FPCS solutions of different concentrations (0.5, 1, 1.5, 2, 2.5 mg/mL) were passed through the column at 1 mL/min. The absorbance (A_1_) was determined, then ethanol solutions of different concentrations (55%, 65%, 75%, 85%, 95%) at different rates (1, 2, 3, 4, 5 mL/min) were selected for elution and the absorbance (A_2_) was measured. To evaluate the best purification conditions, the adsorption rate, adsorption capacity, resolution rate, desorption capacity and recovery were calculated as follows:Adsorption rate (%) = (A_0_ − A_1_)/A_0_ × 100%;
Adsorption capacity = (A_0_ − A_1_) × V_0_/M;
Desorption rate (%) = A_2_/(A_0_ − A_1_) × 100%;
Desorption amount = A_2_ × V/M;
Recovery (%) = C_α_V_α_/C_β_V_β_ × 100%
where M is the mass of resin; A_0_ is the absorbance value of pigment stock solution; V_0_ is the volume of pigment stock solution and V is the volume of desorption solution; A_1_ and A_2_ are the absorbance values of the supernatant after adsorption and desorption, respectively; C_α_ is the pigment concentration in the recovery solution; V_α_ is the volume of recovered solution; C_β_ is the pigment concentration of the injection solution; and V_β_ is the injection volume.

### 2.4. Color Value of FPCS

The color value of FPCS was determined according to the Chinese standard GB15961-2005 by accurately weighing 0.05–0.1 g (accurate to 0.0002 g) pigment into a 50 mL beaker and dissolving the pigment in distilled water, then transferring it to a 100-mL vessel and shaking well. Absorbance was measured at 370 nm in a 1-cm cuvette and the color value of the pigment was calculated according to the following formula:$$E_{1\mathrm{cm}}^{1\%}(370\;\mathrm{nm})=A\;\times \;n/m\;\times \;1/100$$
where *E* (370 nm) represents the color value of the sample, 1% is the sample concentration and 1 cm is the cuvette size; A is the absorbance value of the sample solution; n is the dilution ratio; and m is the mass of the sample in grams.

### 2.5. Antioxidant Capacity of FPCS

Using the Lambert–Beer Law, various volumes of a 1.0 mg/mL FPCS solution (0.25, 0.4, 0.55, 0.7, 0.85 and 1 mL) were mixed in 10 mL volumetric flasks with 2.5 mL of 2,2-diphenyl-1-picrylhydrazyl (DPPH) solution. The volume was adjusted with absolute ethanol, shaken well and the absorbance was measured. Then antioxidant activity, represented by DPPH clearance, was calculated using the Lambert–Beer law according to the formula:W% = A_1_ − A_s_/A_1_ × 100%
where W represents DPPH clearance; A1 represents absorbance of the DPPH solution; and A_s_ is the absorbance of DPPH solution in the presence of the sample. Clearance of the free radical, 2,2′-azinobis-(3-ethylbenzthiazoline-6-sulphonate (ABTS), was measured in a similar way. IC_50_ is given by the W value when the clearance rate is 50% according to the formula above.

### 2.6. Mice

Male and female Kunming (KM) mice [19] (Charles River, Beijing, China) were used in this study. Mice were housed in a climate-controlled room (25 °C) on a 12 h light–dark cycle (lights on at 8:00 a.m.) with ad libitum access to standard rodent chow. All animal experiments performed were approved by the Animal Care and Use Committee of Guangzhou University and followed the guidelines published in the National Institutes of Health Guide for the Care and Use of Laboratory Animals.

### 2.7. Acute Toxicity Test

Based on Horn’s method [20,21], thirty KM mice, which are often preferred for laboratory research on vaccines or drugs [22,23,24,25], were used. All animals were housed in a standard environment (12 h light/dark cycle, 25 ± 1 °C) and had free access to a standard rodent chow and fresh water. The mice were adaptively raised for seven days before the start of the actual experiment. Both male and female mice of 18–22 g weight were used. Mice were randomly divided into five groups with six mice in each group. After 48 h of normal feeding, mice were subjected to fasting for 12 h with freely available drinking water. Intragastric doses of 6.0, 3.25, 1.75 and 0.50 g/kg FPCS, dissolved in saline at 10 g/0.4 mL, were delivered by gavage to the four experimental groups. Vehicle-treated mice were used as a control group. The gavage volume used in this experiment was 10 g/0.4 mL. After gavage, mice were given ordinary pellet feed and drinking water. The clinical symptoms and the death of mice were recorded for one week. The acute toxicity classification was based on the LD_50_ (GB15193.3-2014). Experimental procedures were approved by the Institutional Animal Care and Use Committee at Guangzhou University and South China Normal University. The methods were carried out in compliance with the accepted recommendations published by the National Institutes of Health Guide for the Care and Use of Laboratory Animals.

### 2.8. FPCS and LPS Treatment

To confirm the antioxidant activity of FPCS in lipopolysaccharide (LPS)-challenged mice, purified FPCS was injected intraperitoneally (i.p.) at doses of 10 mg/kg, 30 mg/kg and 90 mg/kg daily for seven days [26]. LPS from *Escherichia coli* serotype O111:B4 (Sigma- Aldrich Shanghai Trading Co., Ltd., Shanghai, China) was diluted in saline and injected into mice at a dose of 0.83 mg/kg [27]. Saline (indicated as Ctrl in the text) or LPS was injected 3 h prior to experimental measurements. Animals were randomly divided into the following six groups:

Group 1—Vehicle control (saline i.p) for 7 days;Group 2—Vehicle for 7 days and LPS (0.83 mg/kg i.p.) on the 8th day;Group 3—FPCS (10 mg/kg i.p.) for 7 days and LPS (0.83 mg/kg i.p.) on the 8th day;Group 4—FPCS (30 mg/kg i.p.) for 7 days and LPS (0.83 mg/kg i.p.) on the 8th day;Group 5—FPCS (90 mg/kg i.p.) for 7 days and LPS (0.83 mg/kg i.p.) on the 8th day;Group 6—FPCS (10 mg/kg i.p.) for 7 days.

The animals used for western blotting were killed by cervical dislocation after 3 h saline or LPS challenge.

Below is the table showing the chemicals (Damao Chemical Reagent Factory, Dongguan, China) used in this study.



### 2.9. Western Blotting

Mouse brains were rapidly dissected on ice and hippocampus tissues were homogenized in a lysis buffer (50 mM Tris pH 7.5, 150 mM NaCl, 5 mM EDTA pH 8.0, 1% SDS and protease inhibitors (Complete Mini; Roche, Basel, Switzerland). After centrifugation at 4 °C (14,000 rpm for 10 min), cellular debris was removed, and the supernatant was collected for western blotting. Tissue lysates were analyzed by sodium dodecyl sulfate-polyacrylamide gel electrophoresis (SDS-PAGE) and separated proteins were transferred to nitrocellulose membranes. Membranes were then blocked with 5% defatted milk in Tris-buffered saline with Tween 20 (TBST) for 1 h and incubated overnight at 4 °C with the following specific primary antibodies against iNos (Abcam, ab178945; dilution 1:1000), nitrotyrosine (Santa Cruz: sc-32757; dilution 1:1000) and Hsp60 (Abcam, ab46798; dilution 1:2000). Anti-GAPDH antibody (AF0006, Beyotime, Jiangsu, China) was used as a loading control. After three washes with TBST, an HRP-labeled secondary antibody (CWS, Taizhou, China) was added at room temperature for 1 h using 5% milk in TBST, followed by three additional washes with TBST. The Immobilon ECL western system (Millipore, Burlington, USA) was then used to visualize the bands, which were quantified and analyzed with Gel-Pro Analysis software (Media Cybernetics, Rockville, MD, USA).

### 2.10. Statistical Analysis

Prism 8.0 for Windows (GraphPad, San Diego, CA, USA) and OriginPro 2020 (OriginLab, Northampton, MA, USA) software was employed for graphing and the statistical analysis of all single factor tests. Design-Expert 12.0 (StAt-Ease Company, Minneapolis, MN, USA) was used to design and analyze the mathematical model of the RSM experiment. A two-sample *t*-test was used for statistical analyses between two-group comparisons. For western blotting analysis, One-Way ANOVA with Post-Hoc test was performed. *p* < 0.05 was considered statistically significant. The data are presented as mean ± SEM.

## 3. Results

### 3.1. Optimization of FPCS Extraction by Response Surface Methodology

The extraction process for FPCS was optimized as an improvement on the traditional solvent extraction method reported previously [28]. Response surface methodology (RSM) is commonly used to optimize complex extraction procedures as it can simultaneously evaluate the effects of different parameters, as well as their interactions [5,29,30]. Using this technique, the effects of ethanol concentration, SL ratio, temperature and time on the extraction rate of FPCS were first determined in single-factor experiments. The highest pigment extraction rate was obtained when the ethanol concentration was 75% (Figure 1A). Higher concentrations of ethanol (>75%) resulted in a decreased extraction rate. The SL ratio affects the solubility of the raw materials in the extraction solvent [31]. A low SL ratio results in a rapid increase in the liquid-phase concentration, thus reducing the solubility of the raw materials. On the other hand, when the SL ratio is high, the raw materials can be fully dissolved, but this may also result in the dissolution of other substances. Therefore, it is crucial to select the appropriate SL ratio by performing a single-factor experiment. The FPCS extraction rate was highest when the ratio was 1:70 (Figure 1B), which showed a significant difference with adjacent values (*p* < 0.05). Temperature is also a very important factor in the extraction process [31,32], which was confirmed by our results for FPCS. Below 70 °C, the extraction rate was positively correlated with temperature. Thereafter, the effect of temperature plateaued, suggesting that 70 °C is optimal for FPCS extraction (Figure 1C). Another crucial factor in the extraction process is time [33,34]. The FPCS extraction rate increased slightly between 15- and 45-min extraction time, and reached its highest level at 45 min (*p* < 0.05), but decreased with longer times (Figure 1D). This suggests that 45 min is the optimal time for FPCS extraction. Indeed, longer times could result in the extraction of other substances, thereby decreasing the relative extraction rate of FPCS.

Using the Box–Behnken experimental design, the four factors of ethanol concentration, SL ratio, extraction temperature and time were used as the test factors for the response surface analysis, with the pigment extraction rate as the reference index, resulting in the optimization of the extraction process (Table 1). The corresponding quadratic equation model was obtained using Design Expert 12.0 software as follows:Y = 48.46 − 1.63A + 0.6083B − 1.9C − 1.00D − 2.35AB − 0.5AC − 1.12AD − 2.32BC − 1.1BD + 1.12CD − 9.81A^2^ − 10.11B^2^ − 7.25C^2^ − 6.5D^2^.

Using analysis of variance (Table 2), it can be seen that the F value of the model was 9.00 and the *p*-value was less than 0.0001, indicating that the model was highly significant, fit the real situation very well, and thus could be used to predict the extraction rate.

### 3.2. Interpretation of Response Surface Models

Response surfaces were plotted to study the effects of the parameters and their interactions on the extraction of FPCS. The fitting function produced three-dimensional response surface plots simultaneously showing the effect of two factors on the pigment extraction rate (Figure 2). Thus, Figure 2A–C shows the response surface plots of the effect of the ethanol concentration (65–85%) together with the SL ratio (1:60–1:80) (Figure 2A), temperature (60–80 °C) (Figure 2B) or extraction time (30–60 min) (Figure 2C). Clearly, ethanol concentration had a significant impact: the FPCS extraction rate increased slightly, then decreased with increasing ethanol concentration. At a given ethanol concentration, SL ratio and time, the extraction rate first went up slightly when the extraction temperature was increased (in the given range), suggesting that temperature does have a positive influence on the FPCS extraction rate. However, the extraction rate went down when the temperature was further increased (Figure 2B,D,F). Thus, a proper temperature is critical in FPCS extraction. The interaction effect, i.e., extraction time with concentration, SL ratio with temperature, on the extraction rate is also shown (Figure 2C,E,F). For example, in the 30–60 min range, time had a significant effect on the FPCS extraction rate, which first increased and then decreased.

The regression model within the Design Expert 12.0 software package was used to optimize the levels of the above significant factors. Taking the pigment extraction rate as the investigation index, the optimized extraction conditions are shown in Table 3. For practical application, the conditions were rounded as follows: ethanol concentration, 76%; SL ratio, 1:75; temperature, 66 °C; and time, 39 min. Under these conditions, experimental validation was carried out with three repeats. The averaged extraction rate was 43.2% and the prediction accuracy was 95.68%. These results suggest that there was no significant difference between the measured average value and the predicted value, i.e., the fitting of the model was good and the prediction was accurate.

### 3.3. Purification of FPCS by Macroporous Resin

#### 3.3.1. Static Adsorption Experiment

To obtain high-purity pigment products and to expand the application range of such pigments, it is often necessary to further purify crude preparations of the pigment and remove impurities [35,36]. Using a macroporous resin as an adsorbent, the effects of different factors on the static and dynamic adsorption and desorption of FPCS were investigated. This allowed the optimal process for resin purification of pigment to be determined and provided a theoretical basis and some scientific data for the production and use of the pigment [37,38,39].

Based on the UV-vis spectrum obtained by full-band scanning (Figure 3A), FPCS had an absorption peak at 370 nm; this wavelength was used subsequently to assay pigment solutions. Three macroporous resins, D101, A8 and HPD100, were selected for static and dynamic adsorption experiments, and their absorbance was measured. The absorbance and pigment concentrations were transformed using a linear regression equation (Figure 3B). In the static test, the adsorption rate with HPD100 was flat after 6 h, with A8 after about 7 h and with D101 after about 10 h. Overall, D101 gave a higher adsorption rate than A8 or HPD100 (Figure 3C). In the static test, the desorption rates of D101 and HPD100 were significantly higher than that of A8, but there was no significant difference between D101 and HPD100 (Figure 3D). Taking above both results into account, the D101 macroporous resin was selected for purifying FPCS.

#### 3.3.2. Dynamic Adsorption Experiment

The best loading volume to use with resin D101 was investigated (Figure 4A). The FPCS concentration in the effluent reached 0.124 mg/mL when the upper column volume was 60 mL. The FPCS concentration was 10% in the initial sample solution as the leakage point of D101; therefore, the upper column volume could be maintained at 60 mL. The adsorption capacity of D101 gradually became saturated as the sample solution volume was increased. However, when the sample volume reached 210 mL, the FCPS in the effluent concentration was 1.0 mg/mL, very similar to that of the sample solution, indicating that the saturation point of D101 had been reached [39,40].

Using the optimum loading volume determined above, we examined the elution of FCPS from the D101 column (Figure 4B). The FPCS concentration in the eluent reached a maximum value at a total eluent volume of 25 mL. However, as the volume of eluent increased further, the concentration of the pigment in the eluent gradually decreased: when the eluent volume reached 50 mL, the FPCS content decreased to its minimum value in the eluent (<0.01 mg/mL), indicating that all the pigment adsorbed by the macroporous resin had been eluted. Considering that the elution peak was single, symmetrical and sharp, the obvious tailing phenomenon showed that the pigment could be completely eluted by 50 mL of 95% ethanol.

Having determined the appropriate loading and elution volumes for FPCS, the eluent concentration and elution rate were analyzed by single-factor analysis. The elution rate was determined by comprehensively considering the production cycle and recovery rate. After purification by D101, the yield of FPCS was 5.73%, and the color value was 29.41. The color value of the pigment increased nearly three-fold after purification. In the low mass concentration range, the resin adsorption capacity and recovery rate increased with increasing the FPCS concentration up to a concentration of 1.50 mg/mL, at which the resin reached its maximum adsorption capacity, remaining essentially unchanged thereafter (Figure 4C). However, when the FPCS concentration was greater than 1.50 mg/mL, recovery decreased because the sample concentration exceeded the adsorption saturation point of the resin. Together, the results suggest that an initial FPCS concentration of 1.50 mg/mL is appropriate. It is known that compounds such as polyphenols and compounds with glycosidic bonds show a weak polarity in aqueous solutions and are easy to elute from macroporous resins using ethanol [41]. We found that the amount of ethanol was critical: at a volume fraction of ethanol ≤ 75%, both the desorption rate and the recovery rate gradually increased, while at a volume fraction > 75% ethanol, there was a downward trend in the recovery rate (Figure 4D). An elution rate of 1.0 mL/min gave the best pigment recovery rate, with higher flow rates resulting in decreased desorption and recovery rates (Figure 4E).

### 3.4. The Stability of Purified and Unpurified FPCS

Plant extracts are an incredible source of key secondary metabolites including polyphenolics, alkaloids and terpenes, having significant biological activity. Growing demand for these phytochemicals requires a suitable method to efficiently extract and process various plant extracts. An important part of the quality control of natural pigment preparations is the assessment of the chemical stability of a final product during the storage period [42]. Moreover, the existence of enzymes such as glycosidases, esterases or oxidases plays a crucial role in the breakdown of secondary plant metabolites [43,44]. Pigments are sensitive to light which can change the functional groups in the pigments and affect their stability. High concentrations of metal ions interact with pigments, resulting in precipitation. Natural pigments are sensitive to light which can change the functional groups in the pigments and affect their stability; thus, most need to be stored in the dark [45]. Therefore, we next tested the sensitivity of FPCS to light. After one week of direct exposure to sunlight, the pigment retention rate was only 72.11% (Figure 5A). In contrast, natural light, which was indoor lighting in this study, decreased the retention rate by about 10% in a week, suggesting it had relatively little effect on the stability of FPCS (Figure 5B). Under dark conditions, the pigment stability was high, with FPCS retention rates after one week at 98.23% and 94.12% for unpurified and purified pigments, respectively (Figure 5C). Therefore, direct sunlight clearly had a detrimental effect on the stability of FPCS.

Sweeteners have also been shown to affect the stability of pigments [46,47]. The present study shows that a low concentration of glucose or sucrose had little effect on the stability of purified FPCS, with FPCS absorbance changing only slightly during storage. However, increasing concentrations of these two sugars had a negative effect: a significant difference (*p* = 0.0122) was observed between the experiments with 2% and 10% glucose, for example. In other words, FPCS is relatively stable in the presence of low concentrations of sucrose and glucose, but high concentrations have a marked impact on the stability of the pigment (Figure 5D,E).

Weak alkaline substances affect the overall pH of the solution, thereby changing the absorbance of the pigments and even achieving color enhancement [48]. Various edible acids are often used as food additives [49]. Therefore, the effects of citric acid and ascorbic acid on the stability of *C. speciosa* pigment were determined in this study. Even at a low concentration of 0.01%, both acids significantly decreased the levels of FPCS after 96 h, with residual levels of 82.65% (*p* = 0.0007) (Figure 5F) and 77.76% (*p* = 0.0047) (Figure 5G) for citric acid and ascorbic acid, respectively. Therefore, ascorbic acid, citric acid and other edible acids should be avoided during pigment storage.

Temperature can also affect the stability of pigments [50]. We found that when FPCS was stored either in a fridge (0–4 °C) or at room temperature (about 20 °C), there was no color change over a period of 15 days, with the retention rate of the pigment remaining above 97% (Figure 5H). This suggests that refrigeration and room temperature had no significant effect on the stability of FPCS, but after being stored at 40–80 °C for 5 h, the absorbance of the pigment decreased slightly, although the retention rate was still greater than 95%. However, when the temperature was increased to 100 °C, the retention rate decreased significantly compared to storage at 40 °C (*p* = 0.01185) (Figure 5I). It is likely that a high temperature might cause the denaturation of some chromoprotein structures of the pigment [51], and thus change the stability of FPSC. Therefore, these results suggest that high temperature storage should be avoided.

### 3.5. Impact of Oxidizing and Reducing Compounds, Preservatives and Metal Ions on the Stability of FPCS

The effect of redox-active compounds on the stability of FPCS was investigated next. When the concentration of H_2_O_2_, a powerful oxidant, was low (0.005%), the pigment was relatively stable, but above 0.05% H_2_O_2_ the retention rate decreased significantly compared to 0.005% H_2_O_2_ (*p* = 0.0029) (Figure 6A), indicating that FPCS may have a degree of antioxidant capacity. The reducing agent, Na_2_SO_3,_ had little effect on the stability of the pigment at 0.5%, but, remarkably, lower concentrations of Na_2_SO_3_ (e.g., 0.005%) reduced its stability (*p* = 0.00209) (Figure 6B).

Preservatives, the most common food additives, inevitably come into contact with pigments [52]. Various concentrations of sodium benzoate and potassium sorbate, for example, are known to affect the stability of pigments [53,54]. We found that sodium benzoate across a range of concentrations had no effect on FPCS, with the retention rate being statistically unchanged (Figure 6C). In the case of potassium sorbate, increasing concentrations gave an apparent increase in the retention rate of FPCS, with 1% potassium sorbate significantly increasing the retention rate over 96 h (*p* = 0.00119) (Figure 6D). Therefore, it seems that potassium sorbate has a protective effect against the fading of FPCS.

Metal ions can also affect the stability of pigments [55], but we observed no significant effect of Na^+^ and Mg^2+^ on the stability of the FPCS. In contrast, Al^3+^, Fe^3+^ and Cu^2+^, at concentrations less than 0.01 M, degraded the pigment with time (Figure 6E). Moreover, when the concentration was greater than 0.01 M, all three ions induced a brownish-red precipitation of the pigment, which may have been due to the formation of a complex between, or a reaction product of, the pigment and the metal ions [56].

### 3.6. Antioxidant Activity of FPCS In Vitro

The assessment of antioxidant capacity is well established and we therefore next used an in vitro evaluation model to determine whether FPCS had antioxidative capacity. Many in vitro evaluation methods are available, such as spectrophotometry, chemical fluorescence and electron spin resonance, to detect and quantitate free radicals [57,58]. Here, we used a well-known spectrophotometric assay involving the neutralization of the free-radical compounds, DPPH radical and ABTS^+^ radical.

#### 3.6.1. Ability to Scavenge DPPH

The DPPH radical clearance rates with FPCS concentrations of 25, 40, 55, 70, 85 and 100 µg/mL were 35.78%, 41.06%, 54.88%, 61.38%, 73.98% and 81.30%, respectively (Figure 7A). It can be seen from the figure that the higher the pigment concentration, the higher the scavenging rate of free radicals and the stronger the antioxidant effect. Curve fitting showed a linear relationship between DPPH radical clearance and pigment concentration in the range tested, with the formula y = 0.634x + 18.437, where R^2^ = 0.9892 (Figure 7A). Using this linear regression equation, it can be calculated that FPCS scavenged the DPPH free radical with an IC_50_ of 49.78 µg/mL.

#### 3.6.2. Ability to Scavenge ABTS^+^

With FPCS concentrations of 25, 40, 55, 70, 85 and 100 µg/mL, the ABTS^+^ radical clearance rates were 33.01%, 39.89%, 45.92%, 56.75%, 66.39% and 72.34%, respectively (Figure 7B). Again, the higher the FPCS concentration, the better the scavenging rate of free radicals and the stronger the antioxidant effect. There was a linear relationship between ABTS^+^ radical clearance rate and FPCS concentration, with the formula y = 0.5466x + 18.219, where R^2^ = 0.9914. Using this linear regression equation, the IC_50_ for FPCS scavenging of the ABTS^+^ free radical was 58.14 µg/mL.

### 3.7. Acute Toxicity Test in KM Mice

The oral LD_50_ of FPCS was determined by Horne’s method [20,21]. KM mice were given oral FPCS at 1.0–10.0 g/kg. There was no significant difference in body weight between the gavage group and the control group after seven days (*p* = 0.713); the mice were in a healthy condition with no adverse reactions, and no deaths were observed. It is worth noting that even at the highest dose, 10 g/kg, FPCS does not cause death in either male or female mice. The acute toxicity analysis of FPCS is summarized in Table 4.

### 3.8. FPCS Prevents LPS-Induced Oxidative Stress in KM Mice

Natural compounds, most of which are derived from plants, have long been known to have antioxidant properties. Based on the in vitro results, we speculated that FPCS might also reduce oxidative stress in vivo. It has been demonstrated that inflammatory mediators such as LPS and cytokines promote the production of inducible nitric oxide synthase (iNOS) in macrophages and microglia [59,60,61]. Nitrotyrosine, a biomarker of oxidative stress [62,63], is generated by the nitration of protein-bound and free tyrosine residues by reactive peroxynitrite molecules. Heat shock protein 60 (HSP60) is another regulator of oxidative stress and a potential modulator of neuroinflammation [64]. We thus evaluated the effect of FPCS on the levels of iNOS, nitrotyrosine and HSP60 in an LPS-induced oxidative-stress mouse model. The results showed that LPS significantly increased iNOS expression in KM mice (*p* = 0.003), but pretreatment with FPCS at 30 mg/kg (*p* < 0.001) and 90 mg/kg (*p* < 0.001) prevented LPS-induced iNOS expression (Figure 8A,B). There was a trend of increased HSP60 expression in LPS-injected KM mice, and pretreatment with FPCS at 10 mg/kg (*p* = 0.006), 30 mg/kg (*p* = 0.001) and 90 mg/kg (*p* = 0.002) significantly reduced LPS-induced HSP60 expression (Figure 8D). Levels of nitrotyrosine remained unchanged among the groups (Figure 8C). These results suggest that FPCS has a number of antioxidant properties.

## 4. Discussion

Due to their supposed medicinal properties in humans, multiple parts of *C. speciosa*, including the seeds, bark and flowers, are used as a type of tea throughout North America and Asia, including China [16]. The present study described the extraction conditions and a quantitative analysis of the floral pigment, FPCS, from *C. speciosa* with further exploration of the potential of the extract as a new source of natural pigment. The antioxidative activity of FPCS in mice was also investigated. Our acute toxicological experiments showed that FPCS was nontoxic. Thus, we believe this study will contribute valuable knowledge about the extraction and biological effects of FPCS, and in particular that FPCS may be a safe source of natural color and may offer protection against oxidative stress damage.

The BBD was used to optimize the effect of the ethanol concentration, SL ratio, extraction temperature and extraction time on the extraction rate of FPCS. The results derived optimal extraction conditions that could be readily applied in a laboratory or production facility. For the purification of the pigment, three macroporous resins (D101, A8 and HPD100) were evaluated, and D101 was chosen due to its higher adsorption rate. The best purification procedure shown below was obtained through the dynamic adsorption single factor test: adding 60 mL pigment solution with a concentration of 1.50 mg/mL to the chromatography column at a flow rate of 1.0 mL/min, followed by 50 mL of wash detergent (75% ethanol) at a flow rate of 2.0 mL/min.

The FPCS yield using a traditional pure-water method was 23.1 ± 0.32% and we were able to improve this to 43.2 ± 0.21% by using ethanol in this study. The study shows that optimal extraction conditions were obtained when the ethanol concentration was 76%, SL ratio 1:75, temperature 66 °C and extraction time 39 min. Pigment stability was evaluated under various conditions, and our results allow us to provide guidance for the storage of FPCS. Briefly, (1) given that high temperature and direct sunlight have a marked impact on its stability, FPCS should be stored in cold storage in the dark. (2) FPCS is a strongly polar water-soluble pigment and remains stable when pH is acidic or neutral but is unstable in an alkaline environment. (3) FPCS is relatively stable in low concentrations of sweeteners, but high concentrations have a negative impact on pigment stability. (4) Edible acids have a significant negative influence on the pigment: both citric acid and ascorbic acid likely degrade FPCS over time. (5) H_2_O_2_ and Na_2_SO_3_ have relatively little effect on the stability of FPCS, indicating that FPCS has a degree of antioxidant and reducing capacity. (6) Sodium benzoate and potassium sorbate have protective effects on FPCS, especially potassium sorbate, which has a certain color increasing effect. (7) FPCS is relatively stable in the presence of Na^+^ and Mg^2+^, but Al^3+^, Fe^3+^ and Cu^2+^ have a significant negative influence on the stability of the pigment.

Worth noting is that our optimization of the extraction process increased the extraction rate of FPCS by about 20% compared to the traditional water extraction method. After purification by macroporous resin, the yield of FPCS was 5.73% and the color value was 29.41, which was nearly 3 times higher than the crude product. In addition, FPCS is not a carotenoid pigment since we could not obtain an extract when using ligroin as the solvent in a preliminary study. Given that we found that the color of FPCS solution became darker accompanied by bubbling when reacting with HCl-Zn, the FPCS may consist of flavonoids. Moreover, FPCS reacted with an FeCl_3_-induced brown solution with precipitation in our preliminary study, indicating it contained phenolic hydroxyl group [51]. The phytochemical analysis of the methanol extract of *C. speciosa* leaves showed the presence of bioactive compounds such as flavonoids, tannins glycosides, phenols, alkaloids, saponnins, steroids and terpernoids [65]. Proximate analysis of *C. speciosa* leaves showed 53% moisture, 2.5% ash, 24% carbohydrate, 12% fats and 8.5% protein [65]. The seed oil of *C. speciosa* contains linoleic acid (28.22%), palmitic acid (19.56%), malvalic acid (16.15%), sterculic acid (11.11%), and dihydrosterculic acid (2.74%) [16]. Together, these studies suggest the functional properties of *C. speciosa* and propose its use as an important bioactive compound.

Our acute in vivo toxicology experiments showed that gavage administration of FPCS is safe in both male and female mice. All mice survived seven days after gavage FPCS and were in healthy condition. Therefore, this study may provide valuable knowledge about the biological effects of FPCS.

The antioxidant activity of FPCS was investigated based on the Lambert–Beer law. This showed that the IC_50_ for the scavenging of DPPH free radicals by FPCS was 49.78 μg/mL, while the IC_50_ for the clearance of ABTS^+^ radicals was 58.14 μg/mL. Thus, FPCS had a scavenging effect on DPPH and ABTS^+^ free radicals, and this antioxidant capacity was concentration dependent. It is known that the ABTS antioxidant activity of *Ceiba speciosa* (A. St.-Hil.) seed oil is IC_50_ = 10.21 μg/mL, which is lower than FPCS. In contrast, the IC_50_ of its DPPH antioxidant activity is 77.44 μg/mL, which is higher than FPCS [16].

Increased oxidative stress has been linked to the development of a variety of diseases in humans, including neurodegenerative disorders, cancer, type 2 diabetes, inflammation, viral infections, autoimmune pathologies and digestive system disorders such as gastrointestinal inflammation and gastric ulcers [66,67,68,69]. Accordingly, the role of reactive oxygen species (ROS) in the etiology of a spectrum of diseases, together with the possibility of a protective effect of natural substances against such ROS, are currently major research topics. Antioxidants are among the most effective natural defenses against disease, offering protection against stressors that might otherwise harm cellular structures. Numerous fruits, vegetables and other plant materials are natural sources of compounds with protective activities, including antioxidant, anticancer, antibacterial, antiviral, antiseptic, anti-inflammatory and anti-allergic compounds [70,71,72,73,74]. Furthermore, the desire to increase fruit and vegetable consumption in the human diet is partially due to the favorable effects of beneficial antioxidants found in these foods on health promotion. Previous research on *C. speciosa* fruit and leaves has revealed anti-inflammatory and antioxidant properties [11,13,14,15], but whether FPCS, in addition to its role as a color additive, has similar activities was unknown before the current study.

Our work revealed that FPCS prevents LPS-induced oxidative stress in mice as indicated by reduced protein levels of iNOS and HSP60. Therefore, the antioxidant activity exhibited by the FPCS extract should be explored for potential pharmaceutical use in the future to elucidate its mechanism of action, and to isolate and characterize its active compounds.

## 5. Conclusions

This study showed that FPCS is nontoxic; therefore, *C. speciosa* flowers are a source of natural pigment that may be used as a color additive, and which can function as an antioxidant. The extension of the proposed production method to other sources of natural products will help to increase the availability and use of natural pigments and should yield a variety of functional compounds with applications in important industrial sectors such as food, cosmetics, medicine and biological components.

The optimal extraction results were obtained under the following conditions: ethanol concentration 76%, SL ratio 1:75, extraction temperature 66 °C and extraction time 39 min. This study increased the extraction rate of FPCS by about 20% compared to the traditional water extraction method. The results regarding the stability of FPCS in different chemical and physical environments may be instructive for its storage and processing.

Moreover, the results derived the optimal conditions for the extraction of FPCS that could be readily applied in laboratory research or a production facility, and future directions should involve the elucidation of its mechanism of action, and the isolation and characterization of its active compounds.

## Figures and Tables

**Figure 1 molecules-27-03555-f001:**
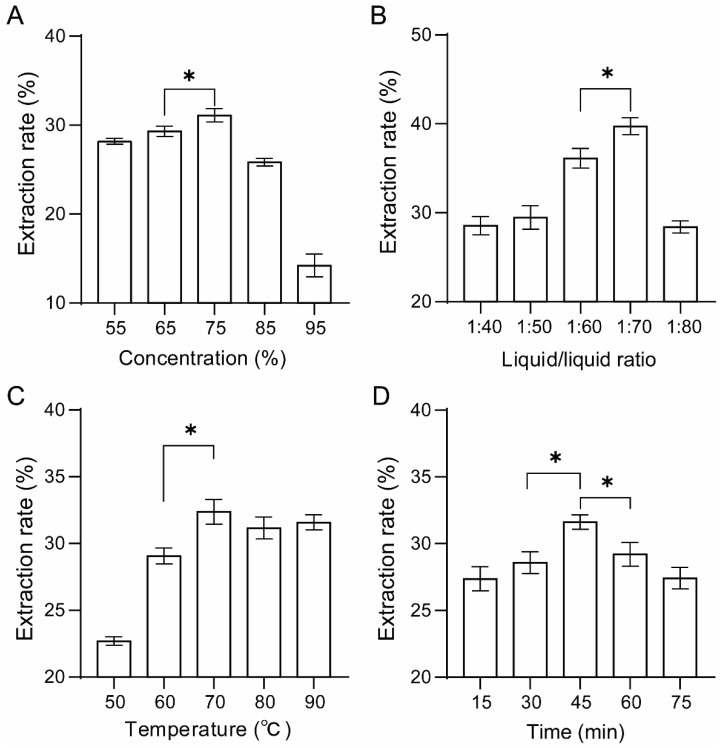
Effects of different extraction conditions on the single-factor extraction rate of FPCS. (**A**) Ethanol concentration. (**B**) Solid/liquid ratio. (**C**) Extraction temperature. (**D**) Extraction time. An asterisk (*) indicates statistical significance at *p* < 0.05.

**Figure 2 molecules-27-03555-f002:**
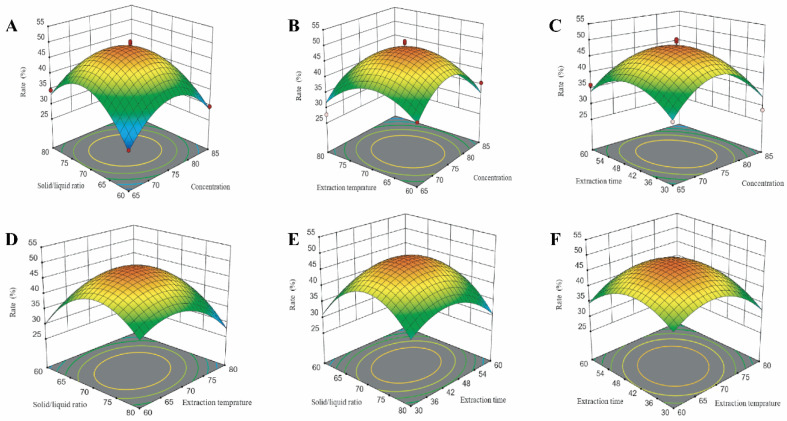
The 3D response surface (RSM) analysis of the extraction rate of FPCS, showing the interactive effects of parameters including (**A**) Solid/liquid ratio and ethanol concentration. (**B**) Extraction temperature. (**C**) Extraction time. (**D**) Solid/liquid ratio and temperature. (**E**) Solid/liquid ratio and time. (**F**) Extraction time and temperature.

**Figure 3 molecules-27-03555-f003:**
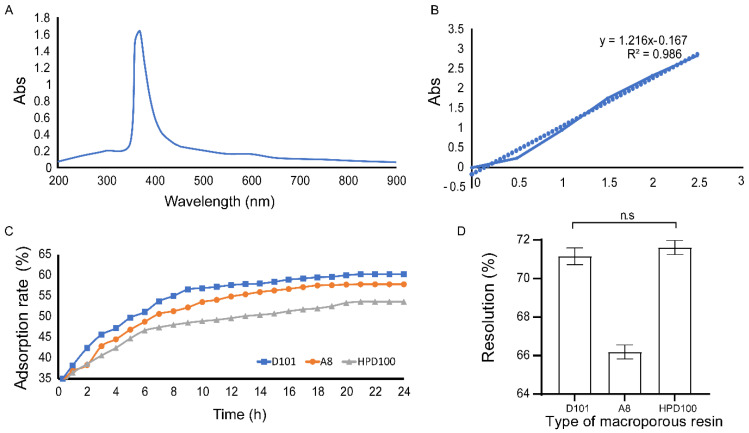
Analysis of the adsorption of macroporous resin. (**A**) The UV-vis spectrum of FPCS showing an absorbance peak at 370 nm. (**B**) Linear regression of pigment concentration and adsorbance at 370 nm. (**C**) Adsorption rate of D101, A8 and HPD100 macroporous resins. (**D**) Higher desorption rates of D101 and HPD100 than A8 macroporous resin.

**Figure 4 molecules-27-03555-f004:**
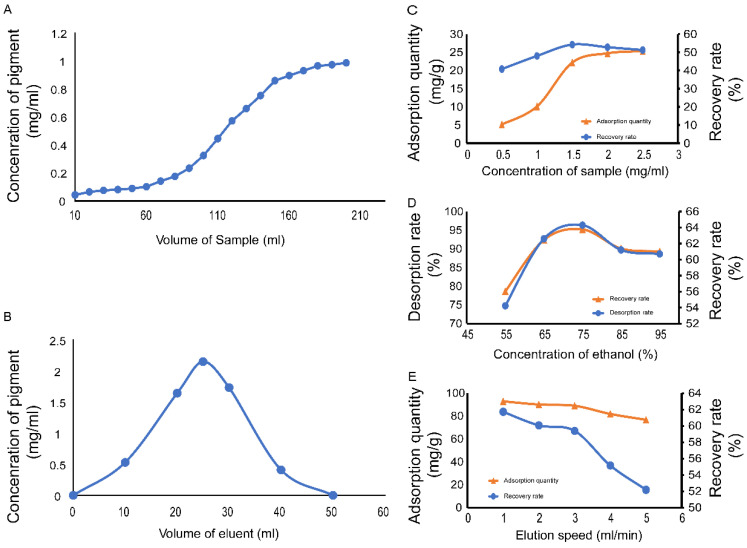
Dynamic breakthrough curve of D101 macroporous resin adsorption. (**A**) D101 leakage curve. (**B**) D101 elution curve. (**C**) Effect of different sample concentrations on the adsorption performance of D101 resin. (**D**) Effect of eluent concentration on the desorption performance of D101 resin. (**E**) Effect of desorption flow rate on the desorption performance of D101 resin.

**Figure 5 molecules-27-03555-f005:**
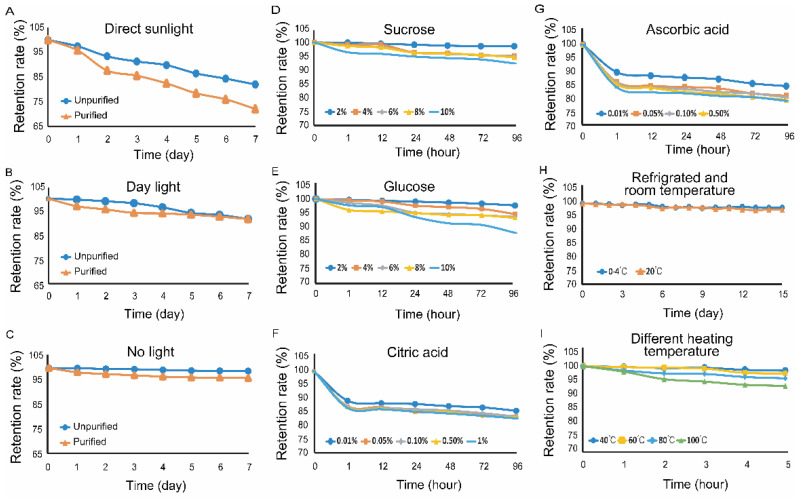
Stability of purified and unpurified FPCS. (**A**) Retention rate of the purified pigment after one week of sunlight irradiation. (**B**) The retention rates of purified and unpurified pigments after one week of natural light irradiation. (**C**) The retention rates of purified and unpurified pigments after storage in the dark. (**D**) The effect of different concentrations of sucrose on the stability of FPCS. (**E**) The effect of different concentrations of glucose on the stability of FPCS. (**F**) The effect of different concentrations of citric acid on the stability of the pigment. (**G**) The effects of different concentrations of ascorbic acid on the stability of the pigment. (**H**) The retention rate of the pigment after 15 days of cold storage and at room temperature. (**I**) The effect of different temperatures on the stability of FPCS.

**Figure 6 molecules-27-03555-f006:**
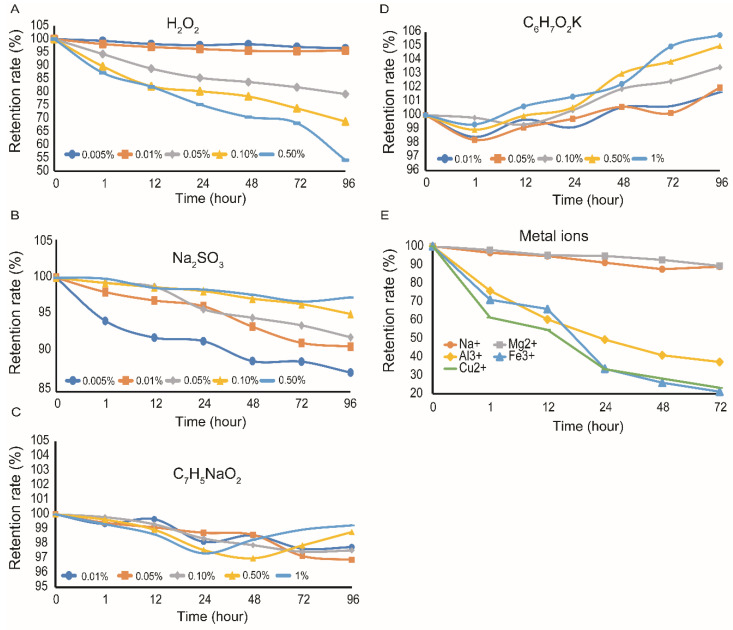
Effects of oxidants and reductants, preservatives and metal ions on the stability of FPCS. (**A**) The effect of different concentrations of H_2_O_2_ on the stability of the pigment. (**B**) The effect of different concentrations of Na_2_SO_3_ on the stability of FPCS. (**C**) The effect of different concentrations of sodium benzoate on the stability of FPCS. (**D**) The effect of different concentrations of potassium sorbate on the stability of the pigment. (**E**) The effect of Na^+^, Mg^2+^, Al^3+^, Fe^3+^ and Cu^2+^ on the stability of the pigment.

**Figure 7 molecules-27-03555-f007:**
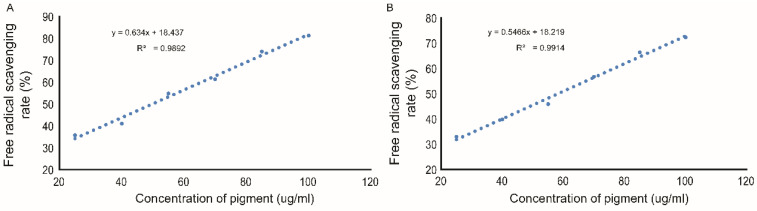
Linear correlation plots of the antioxidant activity of FPCS in vitro. (**A**) DPPH radical scavenging activity of pigment. (**B**) Linear relationship between ABTS^+^ radical clearance rate and pigment concentration.

**Figure 8 molecules-27-03555-f008:**
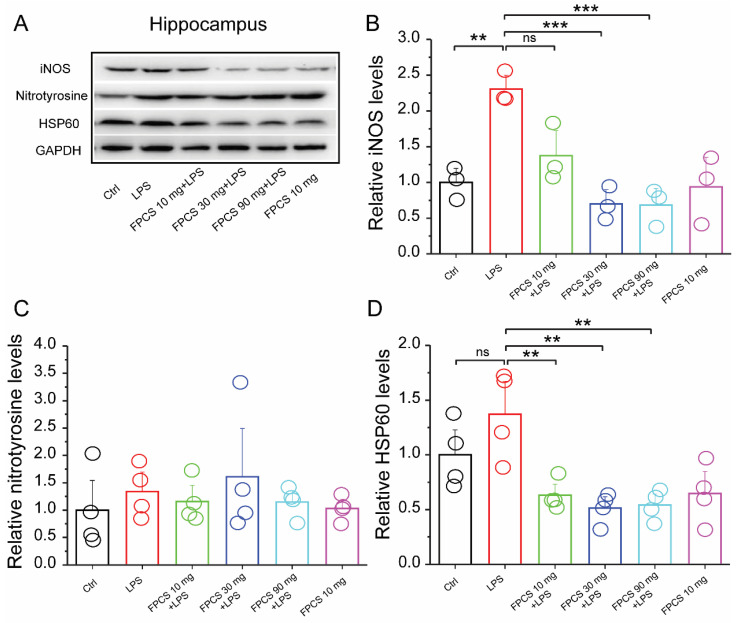
FPCS treatment prevents LPS-induced oxidative stress in a mouse hippocampus. (**A**) Representative immunoblots of mPFC extracts from Ctrl, LPS, 10 mg FPCS + LPS, 30 mg FPCS + LPS, 90 mg FPCS + LPS and 10 mg FPCS alone. (**B**) Quantification of the immunoblots showing that LPS significantly increases iNOS expression, but pretreatment with 30 mg and 90 mg FPCS prevents this. (**C**) Identical expression of nitrotyrosine levels among groups. (**D**) Increased expression of HSP60 in LPS-treated mice is prevented by 10 mg, 30 mg and 90 mg FPCS treatment. One-way ANOVA followed by Tukey’s test for multiple group comparison. Data are presented as means ± SEM. ** *p* < 0.01, *** *p* < 0.001 and ns, not significant.

**Table 1 molecules-27-03555-t001:** Box–Behnken test and effect value.

Number	Concentration (%)	Liquid (g/mL)	Temperature (°C)	Time (min)	Rate (%)
1	65	1:70	70	30	33.2
2	75	1:70	80	30	32.4
3	75	1:80	70	30	38.2
4	75	1:60	70	30	31.8
5	75	1:70	60	30	39.2
6	85	1:70	70	30	28.6
7	65	1:80	70	45	34.6
8	65	1:70	60	45	34.8
9	65	1:70	80	45	27.8
10	65	1:60	70	45	27.6
11	75	1:80	60	45	30.4
12	75	1:70	70	45	48.1
13	75	1:80	80	45	26.9
14	75	1:70	70	45	50.4
15	75	1:70	70	45	49.8
16	75	1:70	70	45	46.7
17	75	1:70	70	45	47.5
18	75	1:60	60	45	28.6
19	75	1:60	80	45	34.4
20	85	1:80	70	45	27.2
21	85	1:70	80	45	26.6
22	85	1:60	70	45	29.6
23	85	1:70	60	45	35.6
24	65	1:70	70	60	35.9
25	75	1:70	80	60	34.9
26	75	1:80	70	60	29.3
27	75	1:60	70	60	27.3
28	75	1:70	60	60	37.2
29	85	1:70	70	60	26.8

**Table 2 molecules-27-03555-t002:** ANOVA analysis of pigment extraction from exotic cotton by response surface analysis.

Variance	Sum of Squares	df	Mean Square	F Value	*p* Value
model	1403.34	14	101.24	9.00	0.0001 **
A-Concentration	31.69	1	31.69	2.84	0.1138
B-Liquid	4.44	1	4.44	0.3986	0.5380
C-Temperature	43.32	1	43.32	3.89	0.0687
D-Time	12.00	1	12.00	1.08	0.3169
AB	22.09	1	22.09	1.98	0.1809
AC	1.00	1	1.00	0.0898	0.7689
AD	5.06	1	5.06	0.4544	0.5112
BC	21.62	1	21.62	1.94	0.1853
BD	4.84	1	4.84	0.4344	0.5205
CD	5.06	1	5.06	0.4544	0.5112
A2	624.66	1	624.66	56.07	<0.0001 **
B2	663.43	1	663.43	59.55	<0.0001 **
C2	341.02	1	341.02	30.61	<0.0001 **
D2	274.12	1	274.12	24.60	0.0002 **
Residual	155.97	14	11.14		
Spurious term	145.52	10	14.55	5.57	0.0562
Pure error	10.45	4	2.61		
Sum	1559.31	28			

** Indicates statistically significant.

**Table 3 molecules-27-03555-t003:** Optimized extraction conditions of FPCS.

	Concentration (%)	Liquid (g/mL)	Temperature (°C)	Time (min)	Rate (%)		
					Theoretical Value	Measured Value	Prediction Accuracy %
Optimized extraction conditions	76.32	1:74.75	66.48	39.44	-	-	-
Adjustment	76	1:75	66	39	45.15	43.2 ± 2.1	95.68

**Table 4 molecules-27-03555-t004:** Measurement of acute toxic effect of FPCS on mice.

Dose	Sex	Number of Animals	Weight		Number of Deaths
			Initial Weight	Final Weight	
10	♂	3	19.8 ± 0.5	31.2 ± 0.6	0
	♀	3	20.2 ± 0.5	31.3 ± 1.0	0
4.64	♂	3	19.9 ± 0.6	30.3 ± 0.9	0
	♀	3	20.1 ± 0.5	30.6 ± 1.8	0
2.15	♂	3	20.5 ± 1.1	31.5 ± 1.3	0
	♀	3	20.3 ± 0.8	30.7 ± 0.7	0
1	♂	3	20.1 ± 1.0	31.8 ± 0.8	0
	♀	3	19.8 ± 0.9	29.6 ± 1.1	0
blank	♂	3	20.8 ± 1.3	30.3 ± 1.4	0
	♀	3	20.6 ± 0.6	31.5 ± 1.3	0

## Data Availability

All data generated or analyzed during this study are included in this article.

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
