# Peer review of "Pigment of Ceiba speciosa (A. St.-Hil.) Flowers: Separation, Extraction, Purification and Antioxidant Activity"

_molecules, 2022, doi:10.3390/molecules27113555_

Round 1

Reviewer 1 Report

Some suggestions for completing the manuscript:

  1. The value of the extraction efficiency of dried plants should be given in the abstract;
  2. What solvent has been used for washing  raw collected flowers;
  3. The exact characteristics of the resins used should be given: porosity, pore size, granulation as well as the name of the supplier;
  4. The breed of laboratory rats should be given

Reviewer 2 Report

All the comments made can be found in the attached file

Reviewer 3 Report

Present research by Chen et al. is focused on the extraction of pigments from Ceiba speciosa flowers. Solid-liquid extraction was applied in the first step and the process was optimized by response surface methodology. Evaluation was followed by purification of pigments by column chromatography and in vitro antioxidant activity and acute toxicity assays were applied for characterization. I have several remarks which have to be improved prior the further consideration.

  • Section 2.1: Voucher specimen number should be added since plant material was identified by a botanist.
  • Section about the chemicals is missing in materials and methods.
  • Section 2.2: Experimental design should be explained in detail. Firstly, you must describe how you prepare one-factor-at-a-time experiments. Further, how did you select factors and their experimental domain for RSM study, why BBD experimental design, etc. Please be thorough and use appropriate references for important claims.
  • Section 2.9: This part is missing statistical treatment of RSM results. Please explain in detail.
  • What about the significant lack of fit? How do you comment and explain this?
  • Lines 271-274: 95% confidence interval is used generally and you say that accuracy was 93.3%. How can you claim that “These results suggest that there is no significant difference between the measured average value and the predicted value”?
  • The highest experimentally observed value of response was 50.4% and you provide optimization with 46.29%, how and why?
  • Discussion is missing the chemical background on Ceiba speciosa Since you did not determine this, you should reflect on the important references.
  • Section 3.4: How is chemical profile of the extracts associated with the stability in different media?
  • Sections 3.6.1. and 3.6.2: You must add term “radical” after DPPH and ABTS+. Correct that throughout the text.
  • Figure 7 is rather useless, you can express results only as IC50 Pay attention that 50 should be in index!
  • Section 4 is generally modest. Please improve it with the discussion on each aspect of the work (extraction, purification, stability, activity and toxicity).
  • Section 5 is too general. Add some specific conclusions obtained directly as a result of your experiments.
